# Glycated Haemoglobin (HbA1c) and Future Physical and Mental Functional Health in the European Prospective Investigation into Cancer (EPIC)-Norfolk Population-Based Study

**DOI:** 10.3390/jpm13091336

**Published:** 2023-08-30

**Authors:** Fiona McLachlan, Alexandra M. Johnstone, Phyo Kyaw Myint

**Affiliations:** 1Ageing Clinical & Experimental Research (ACER) Team, Institute of Applied Health Sciences, University of Aberdeen, Room 4.013, Polwarth Building, Aberdeen AB25 2ZD, UK; 2The Rowett Institute, University of Aberdeen, Aberdeen AB24 3FX, UK; 3Aberdeen Cardiovascular & Diabetes Centre, Institute of Medical Sciences, University of Aberdeen, Aberdeen AB24 3FX, UK

**Keywords:** diabetes, health-related quality of life, epidemiology, functional health, glycated haemoglobin

## Abstract

Little is understood about the relationship between glycated haemoglobin and future functional health in the general population. In this work, we aimed to assess if glycaemic control is associated with future physical and mental functional health at 18-month follow-up in a UK general population, in those with and without diabetes. This work was a cross-sectional study. Between 1995 and 1997, participants of the European Prospective Investigation into Cancer, Norfolk, attended a health check including blood testing for haemoglobin A1c (HbA1c) and completed a health and lifestyle questionnaire. Eighteen months later, self-reported physical and mental functional health were assessed using short form-36 (SF-36). Outcomes of interest included physical and mental component summary (PCS and MCS, respectively) scores of the SF-36. A total of 7343 participants (56% women, mean (SD) 58.1 ± 9.5 years) were eligible to be included, of whom 167 had prevalent diabetes. In our linear regression analysis, a higher HbA1c (mmol/mol) was found to be associated with a poorer PCS score (coefficient −0.15 (*p* < 0.0001)) at follow-up. After adjustment for comorbid conditions, including obesity, this association was no longer statistically significant. A higher HbA1c (mmol/mol) was associated with a better MCS score at follow-up; this finding was significant when adjusted for comorbid conditions (coefficient 0.029 (*p* < 0.05)). Our findings suggest that the association between a higher HbA1c and poorer physical functional health is explained by a higher BMI and comorbidity status in a general population. While higher HbA1c was found to be associated with higher mental functional health at follow-up, the magnitude of this association was small. Healthy responder bias and unmeasured confounding variables may have influenced this result; thus, it should be interpreted with caution.

## 1. Introduction

Diabetes is associated with a high burden of morbidity and mortality and has a rising economic burden [1]. Haemoglobin A1c (HbA1c) is a key tool for the diagnosis and management of diabetes [2], providing a measurement of diabetes control over the two-to-three-month half-life of red blood cells [3]. It is acknowledged that poor glycaemic control has a negative impact on prognosis in people with diabetes, whilst good control of HbA1c at a level of less than 48 mmol/mol (6.5%) has been consistently shown to prevent progression of micro- and macrovascular complications [4,5]. The UK Prospective Diabetes Study found that every 1% reduction in HbA1c was associated with a 37% risk reduction for macrovascular complications and a 21% risk reduction for any endpoint including deaths related to diabetes and all-cause mortality [4]. However, the relationship between glycaemic excursion and health-related quality-of-life (HRQoL) outcomes, in particular for those without diagnosed diabetes, is less clear.

HRQoL is a key outcome in diabetes and related chronic health conditions. Though not specific to diabetes, the short-form, 36-item questionnaire (SF-36) is a well-established tool for investigating self-reported HRQoL [6] and is widely used in diabetes research [7]. Its summary scores, Physical Component Summary (PCS) and Mental Component Summary (MCS), have been shown to be associated with chronic diseases as well as mortality [8,9,10]. Though this may reflect comorbidity burden, it has been proposed that HRQoL acts as an indicator for mortality in individuals with diabetes [11,12]. Previous studies performed in people with diagnosed types 1 and 2 diabetes have presented varying evidence supporting an association between better glycaemic control and better HRQoL [12,13,14,15,16]. Neumann et al. [17] compared SF-36 outcomes between healthy individuals and those with categorised pre-diabetic states, finding participants with pre-diabetes had poorer HRQoL. Limited further review of this relationship exists in the literature, including an assessment of the linear relationship between glycated haemoglobin and SF-36 outcomes, alongside the predictive ability of HbA1c for longer-term HRQoL. 

A large proportion of the adult population have undiagnosed diabetes and pre-diabetes, often resulting in sequalae of diabetes at diagnosis [18]. To date, no studies have investigated the predictive, linear relationship between glycated haemoglobin and HRQoL in people without diagnosed diabetes. Against this background, our study aimed to determine if glycaemic excursion predicts HRQoL in a large, population-based cohort including those with and without a diagnosis of diabetes.

## 2. Materials and Methods

A cross-sectional study was carried out using the Norfolk (UK) cohort of the European Prospective Investigation into Cancer (EPIC-Norfolk); comprehensive details for recruitment and measurement methods for EPIC-Norfolk are described elsewhere [19]. Through collaboration with 35 Norfolk-based general practices, between 1993 and 1997, adults aged 40–79 years were invited to participate by mail. At recruitment end in 1997, 25,639 participants had completed a Health and Lifestyle Questionnaire and attended a health check at which basic measurements and blood samples were taken. Funding for HbA1c testing became available in 1995 [20]; after this, an additional blood sample was taken for HbA1c measurement. 

At initial clinic visit, weight was measured to the nearest 0.1 kg using salter scales in light clothing without socks or shoes and a stadiometer was used to measure height to the nearest 0.1 cm without shoes. The EPIC Health and Lifestyle Questionnaire contained questions regarding age, smoking, physical activity, alcohol use, occupation, and medical history. Self-reported physical activity was categorised into four groups: inactive, moderately inactive, moderately active, and active, as described by Wareham et al. [21]. Alcohol intake was derived from a food-frequency questionnaire collected at initial visit using a compositional analysis from a frequency estimates program [22]. Comorbidity was ascertained by asking “Has the doctor ever told you that you have any of the following?”, followed by a list of medical conditions including diabetes, stroke, and cancer, as well as heart and lung diseases. Further demographic information including the Townsend Deprivation Index was derived from the 1991 census.

HbA1c was measured using high-performance liquid chromatography using a Bio-Rad Diamat system (Richmond, CA, USA). The coefficient of variation was 3.6% at the lower end of the range (mean, 4.9%).

Eighteen months after enrolment, participants were invited by mail to complete a psychosocial questionnaire including SF-36 questions. PCS and MCS were computed following an algorithm that combines the eight subsets of the SF-36, as described elsewhere [23]. Participants with HbA1c measurement at initial visit and complete return of SF-36 questionnaire 18 months later were included in our analysis. Those with missing values for age, sex, and comorbidity were excluded from our analysis, whilst participants with missing values for other covariates were excluded from relevant statistical models. 

### 2.1. Ethical Considerations

Ethical approval was obtained from the Norwich Research Ethics Committee (UK).

### 2.2. Data Availability 

Data are available upon request, with applications made to the EPIC-Norfolk steering committee for consideration.

### 2.3. Statistical Analysis

Statistical analysis was conducted using R, version “Ghost Orchid” [24]. HbA1c was split into quintiles, with the highest quintile reflecting the American Diabetes Association range for pre-diabetes level of HbA1c of 39 mmol/mol (5.7%) and above [25]. Baseline characteristics were tabulated across each quintile of HbA1c, with analysis of variance (ANOVA) or Chi-squared tests performed to compare the characteristics’ differences among HbA1c quintiles. Mean scores for MCS and PCS at 18-month follow-up were tabulated across HbA1c quintiles, again with trend for linearity assessed using the ANOVA method. 

Logistic regression, via backward stepwise method, was used to assess risk factors for participants being in the highest quintile for HbA1c. A general linear regression model was used to examine the relationship between HbA1c with PCS and MCS at 18-month follow-up as continuous variables. Univariate models were initially created, with further variables added in subsequent models to achieve our final model. Our final regression model (Model 4) was derived using backward stepwise regression method with PCS variable. Though this process removed physical activity status, it was re-added as it was considered an important explanatory variable. Separate linear regression models were then created with covariates: Model 1: age and sex; Model 2: Model 1 + smoking status, physical activity status, and alcohol intake; Model 3: Model 2 + body mass index (BMI); Model 4 (as achieved by backward stepwise regression): Model 3 + medical conditions (myocardial infarction (MI), stroke (CVA), cancer (CA), asthma, chronic obstructive pulmonary disease (COPD), diabetes (DM)) + deprivation index. To ensure consistency, these models were used to examine both PCS and MCS. 

## 3. Results

A total of 7343 participants were eligible to be included in the study. They had complete information for age, sex and comorbidity, HbA1c, and SF-36. This included 3230 men (44%) and 4113 (56%) women with an age range of 39–78 years (mean (SD) 58.1 ± 9.5 years) at the baseline. 

Mean (SD) HbA1c was 34.3 ± 9.0 mmol/mol (5.29 ± 0.81%). A total of 288 participants had diabetic-range HbA1c (≥48 mmol/mol or ≥6.5%), with only 163 of them (56.6%) diagnosed with diabetes. A large proportion were found to have pre-diabetic-range HbA1c (range ≥ 39 mmol/mol, <48 mmol/mol or ≥5.7%, <6.5%) (1054, 14.4%). Twenty-four participants with pre-diabetes-range HbA1c had diagnosed diabetes. The samples’ SF-36 PCS mean (SD) was 48.01 ± 9.94 and MCS mean was (SD) 52.11 ± 9.41. 

Demographic and clinical characteristics of participants by HbA1c quintile are presented in Table 1, with the highest HbA1c quintile reflecting individuals with pre-diabetic-and-above-range HbA1. A higher HbA1c quintile was associated with increasing age, male sex, higher BMI, greater deprivation, lower physical activity level, and smoking. Each rising quintile had a higher proportion of participants who abstained from alcohol. Higher prevalence of diabetes, MI, and cancer (borderline significant) was found in the highest HbA1c quintile. The mean PCS at follow-up decreased with each rising quintile, whilst MCS showed a small increase over the quintiles, with these trends being statistically significant. 

Table 2 shows the association between covariates identified via backward stepwise regression that are associated with increased odds of having the highest quintile of HbA1c. Chronic obstructive pulmonary disease (COPD), prevalent stroke, cancer, deprivation index, and physical activity status were removed from the model after backward stepwise regression. The highest quintile HbA1c was found to be significantly associated with increasing age, higher BMI, positive smoking history, and a lower level of alcohol consumption. Participants with the highest quintile HbA1c were also found to be more likely to have a history of MI and asthma, though these associations were weaker. Self-reported physical activity level was not associated with the highest quintile HbA1c. 

Table 3 illustrates the results for linear regression models for HbA1c with PCS and MCS at 18-month follow-up. The univariate model for PCS revealed a significant coefficient of −0.15 (*p* < 0.0001). For every one-unit increase in HbA1c (mmol/mol), the mean PCS decreased by 0.15 points. There was a sharp decrease in both the magnitude of effect and evidence to reject the null hypothesis in Models 3 and 4. There was a small positive correlation between a rising HbA1c and MCS in our univariate model: for every one-unit increase in HbA1c (mmol/mol), the mean MCS increased by 0.056 points (*p* < 0.0001). When adjusted for covariates in Model 3, this association was no longer statistically significant. However, when adjusted for comorbidities and deprivation index in Model 4, this association increased in size and had a borderline significant result (coefficient 0.029, 95% CI 0.0013–0.057).

## 4. Discussion

This study is the first to investigate the association between HbA1c and HRQoL at follow-up in a large population including people without a diagnosis of diabetes. One striking finding is the prevalence of pre-diabetes: some 1030 (14%) study participants were found to have pre-diabetes-range HbA1c in the absence of a diagnosis of diabetes. This is in keeping with similar cross-sectional study findings. Mainous et al. [18] also reported a rising prevalence of pre-diabetes in the adult English population between 2003 and 2011, from 11.6% to 35.3%; this is likely linked to the rise in obesity rates. Ongoing increasing rates of pre-diabetes are concerning, as they could indicate a future healthcare burden.

### 4.1. Physical Component Summary

Higher HbA1c was associated with poorer PCS at follow-up. However, after adjusting for BMI and prevalent comorbidities, this association was no longer statistically significant, suggesting it is likely mediated by these factors. Previous studies have identified a similar relationship in people with diagnosed type 1 and type 2 diabetes. Engström et al. [14] found a weak negative association between rising HbA1c and PCS in 2726 participants with either type 1 or 2 diabetes. This relationship appeared stronger when a high-risk HbA1c group (HbA1c > 8.6%) was examined alone; their findings remained significant when adjusted for covariates. 

The literature underlying this association is inconsistent. Kuznetsov et al. [15] examined 1876 people with type 2 diabetes and also found a small, negative correlation (−0.01 (−0.1, −0.003), *p* < 0.001) between HbA1c and self-reported physical well-being, which was no longer significant in their multivariate models. In a group of 2499 participants with type 2 diabetes, Nicolucci et al. [16] investigated the impact of self-perceived frequency of hyper- and hypo-glycaemic events on the relationship between HRQoL and self-perceived physical well-being. Once adjusted for demographics, comorbidity, and episodes of hyper- and hypoglycaemia, a positive correlation between HbA1c and PCS 0.58 (*p* < 0.001) was identified. The authors suggest that perceived physical function may be influenced by perceived treatment efficacy along with the physical impact of hypo- and hyperglycaemic events. 

Our work is novel in that we examined this relationship in a large, UK, general adult population, including people without diabetes and across a large range of HbA1c. Our univariate analysis demonstrates a correlation estimate of −0.15 (95% CI −0.17–−0.12), representing a small difference in physical well-being in real terms for each unit rise in HbA1c (mmol/mol). The sharp decrease in both the magnitude of effect and evidence to reject the null hypothesis in Models 3 and 4 when BMI and comorbidities were added highlights that the possible mechanism underlying this relationship includes both factors that predispose to diabetes and poor physical function, including age, BMI, and the sequalae of poorly controlled diabetes such as MI and stroke.

### 4.2. Mental Component Summary 

Increasing HbA1c was found to correlate with better MCS at 18-month follow-up in our study. Though the magnitude of effect was small, it remained significant when adjusted for age, and its size and significance level increased when adjusted for comorbidities. These findings contrast what is found in the wider literature for people with diabetes. In a group of 2499 participants with type 2 diabetes, Nicolucci et al. [16] found higher HbA1c to correlate with poorer MCS, though this was no longer significant when hypo- and hyperglycaemic events were adjusted for. These findings were replicated in a cross-sectional analysis of the ADDITION trial [15], and Engström et al. [14] also found a negative association between MCS and HbA1c, which persisted after adjustment for confounding variables.

Our study population was inherently different to those investigated previously. It is possible our findings reflect the inclusion of people without diabetes. Alongside this, our work included a population with a wide age range. Though we accounted for age as a covariate in linear regression, future research exploring effect modification of this association by age group (e.g., young, middle-aged, old) would help explore the influence of phase of life on this finding. Kuz et al. [15] found the largest impact on diabetes-specific quality of life was related to “freedom to eat”—limitations on eating food items that may be perceived to bring joy or feelings of fulfilment. This may help explain our findings in a population that largely was not diagnosed with diabetes. Diabetes-related restrictions on freedom to eat were not imposed. Therefore, perhaps individuals can take joy or enhanced mental well-being from eating carbohydrate-rich foods, with less awareness of the acute effects on blood sugar control or long-term effects on health.

### 4.3. Strengths and Limitations

The exploratory nature of this analysis ensures that multiple hypothesis tests were conducted, without adjustment for multiple testing. Correction methods were considered too conservative. 

As a population-based study, healthy responder bias cannot be ruled out, although EPIC-Norfolk population cohort’s characteristics are similar to other representative samples of the UK, albeit with a lower prevalence of smoking. Though HbA1c is commonly used as a measure of diabetes control, it can be influenced by factors such as anaemias, haemoglobin variants, haemolysis, and uraemia in renal failure [26]. These factors were not included in our analysis; it is unclear to what degree this may have influenced the results. The complex interplay between HbA1c and self-perceived mental functional status at follow-up is likely subject to further sources of confounding. We do not present information on concurrent depression, vascular disease, visual impairment, admission to hospitals related to sugar control (e.g., ketoacidosis), medication use, and further unmeasured factors that may influence the HRQoL. It has also been argued that the SF-36 questionnaire is not in tune with diabetes-related quality-of-life problems [13,27]. 

## 5. Conclusions

This work provides valuable insight into associations with glycaemic control in the general population. To the best of our knowledge, this is the first large-scale study to investigate whether HbA1c predicts HRQoL at 18 months in a general adult population. Rising HbA1c was found to be associated with poorer PCS, though this was no longer significant after adjustment for BMI and comorbidities, suggesting these are important confounders to consider. Our finding of improved mental functional health with rising HbA1c requires cautious interpretation. The magnitude of the effect estimate was small, and this may be influenced by healthy responder bias and unmeasured confounding variables. This work furthers our understanding and highlights areas of future research in the relationship between glycaemic control and health-related quality of life in populations without diagnosed diabetes.

## Figures and Tables

**Table 1 jpm-13-01336-t001:** Baseline characteristics and SF-36 outcomes by HbA1c quintile.

	1st	2nd	3rd	4th	5th	*p*-Value
HbA1c range (mmol/mol)	≤29	29.1–32.2	32.3–35.5	35.6–38.8	≥38.8	
HbA1c range (%)	≤4.80	4.81–5.10	5.11–5.40	5.41–5.69	≥5.70	
Number	1822	1408	1599	1172	1342	
**Demographic**						
Male (%)	739 (40.6)	607 (43.1)	711 (44.4)	535 (45.6)	638 (47.5)	0.0016
Female (%)	1083 (59.4)	801 (56.9)	888 (55.6)	637 (44.4)	704 (52.5)	
Age mean (sd)	52.2 (9.2)	56.8 (9.4)	58.2 (9.3)	60.4 (8.9)	62.7 (8.5)	<0.0001
BMI mean (sd) m^2^	25.6 (3.6)	25.7 (3.6)	26.2 (3.8)	26.3 (4.0)	27.31 (4.4)	<0.0001
Townsend Deprivation Index mean (sd)	−2.3 (2.1)	−2.3 (2.1)	−2.2 (2.2)	−2.1 (2.2)	−2.0 (2.3)	0.00056
**Activity level**						<0.0001
Inactive (%)	446 (24.5)	389 (27.6)	438 (27.4)	380 (32.4)	519 (38.7)	
Moderately inactive (%)	534 (29.3)	418 (29.7)	488 (30.5)	342 (29.2)	348 (25.9)	
Moderately active (%)	462 (25.4)	327 (23.2)	383 (24.0)	267 (22.8)	266 (19.8)	
Active (%)	380 (20.9)	274 (19.5)	290 (18.1)	183 (15.6)	209 (15.6)	
**Smoking status**						<0.0001
Current smoker (%)	148 (8.2)	115 (8.2)	169 (10.7)	141 (12.1)	160 (12.0)	
Ex-smoker (%)	706 (38.9)	563 (40.2)	657 (41.4)	502 (43.1)	619 (46.3)	
Never smoker (%)	960 (52.9)	721 (51.5)	760 (47.9)	523 (44.9)	559 (41.8)	
**Alcohol intake** Units per week						
0 (%)	179 (9.9)	145 (10.4)	171 (10.8)	146 (12.4)	207 (15.6)	<0.0001
>0–7 (%)	906 (49.9)	705 (50.4)	837 (52.7)	642 (54.6)	698 (52.0)	
>7–<14 (%)	376 (20.7)	299 (21.4)	318 (20.0)	210 (17.9)	233 (17.5)	
14–21 (%)	173 (9.5)	125 (8.9)	145 (9.1)	87 (7.4)	100 (7.5)	
>21 (%)	183 (10.1)	124 (8.9)	116 (7.3)	79 (6.7)	90 (6.8)	
**Prevalent Illness**						
Diabetes diagnosis (%)	5 (0.3)	6 (0.4)	4 (0.3)	9 (0.8)	143 (10.7)	<0.0001
Diabetes drugs (%)	0 (0)	0 (0)	2 (0.1)	5 (0.4)	123 (9.2)	<0.0001
Stroke (%)	25 (1.4)	12 (0.9)	17 (1.1)	16 (1.4)	28 (2.9)	0.057
MI (%)	27 (1.5)	25 (1.8)	37 (2.3)	35 (3.0)	67 (5.0)	<0.0001
Cancer (%)	88 (4.8)	81 (6.1)	97 (6.1)	77 (6.6)	99 (7.4)	0.045
Asthma (%)	172 (9.4)	113 (8.0)	132 (8.3)	103 (8.8)	133 (9.9)	0.35
COPD (%)	138 (7.6)	114 (8.1)	146 (9.1)	118 (10.1)	122 (9.1)	0.59
**SF-36 Outcomes at follow-up**						
PCS mean (sd)	49.6 (9.3)	48.9 (9.3)	48.1 (9.7)	46.8 (10.4)	45.7 (10.9)	<0.0001
MCS mean (sd)	51.2 (9.4)	52.2 (9.3)	52.1 (9.3)	52.5 (9.6)	52.9 (9.5)	<0.0001

Data presented in number and percent (%) or mean and standard deviation unless otherwise stated; *p*-value calculated using Chi-squared or ANOVA method. BMI: body mass index; MI: myocardial infarction; COPD: chronic obstructive pulmonary disease. SF-36 PCS: short-form 36 physical component summary; SF-36 MCS: short-form 36 mental component summary.

**Table 2 jpm-13-01336-t002:** Risk factors for highest quintile HbA1c.

	OR	95% CI	*p*-Value
**Demographic**			
Sex			
Female	1		0.072
Male	1.13	(0.99–1.29)	
Age	1.06	(1.06–1.07)	<0.0001
BMI	1.08	(1.06–1.09)	<0.0001
**Medical History**			
MI			
No previous MI	1		0.019
Previous MI	1.47	(1.07–2.02)	
Asthma			
No asthma	1		0.036
Asthma	1.26	(1.02–1.55)	
**Lifestyle**			
Smoker			
Never	1		<0.0001
Former	1.10	(0.95–1.26)	
Current	1.80	(1.46–2.22)	
Physical activity			
Active	1		0.36
Moderately Active	0.92	(0.75–1.13)	
Moderately Inactive	0.88	(0.72–1.07)	
Inactive	1.00	(0.83–1.22)	
Alcohol intake			
None	1		0.0036
>0–7	0.83	(0.69–1.00)	
>7–14	0.71	(0.57–0.89)	
>14–21	0.68	(0.51–0.90)	
>21	0.61	(0.45–0.82)	

Logistic regression model for highest quintile HbA1c. Each covariate estimate was adjusted for the others. Factors remaining after stepwise regression process with physical activity status re-added. BMI: body mass index; MI: myocardial infarction. Alcohol intake in units per week.

**Table 3 jpm-13-01336-t003:** Linear regression models for HbA1c (mmol/mol) by SF-36 summary outcome.

	SF-36 PCS	SF-36 MCS
Univariate	−0.15	0.056
	(−0.17–−0.12) ***	(0.032–0.080) ***
Model 1	−0.069	0.0044
	(−0.094–−0.044) ***	(−0.020–0.029)
Model 2	−0.054	0.011
	(−0.079–−0.029) ***	(−0.014–0.035)
Model 3	−0.031	0.012
	(−0.055–−0.0060) *	(−0.012–0.037)
Model 4	−0.011	0.029
	(−0.039–0.016)	(0.0013–0.057) *

Data presented as coefficient estimate and 95% confidence interval. SF-36 PCS: short-form 36 physical component summary; SF-36 MCS: short-form 36 mental component summary; *: *p* < 0.05, ***: *p* < 0.0001. Model 1: age and sex. Model 2: Model 1 + smoking status, physical activity status, alcohol intake. Model 3: Model 2 + body mass index. Model 4: Model 3 + comorbidity (diabetes, myocardial infarction, stroke, cancer, asthma, chronic obstructive pulmonary disease) + Townsend Deprivation Index.

## Data Availability

Data are available upon request, with applications made to the EPIC-Norfolk steering committee for consideration.

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
