# Peer review of "Glycated Haemoglobin (HbA1c) and Future Physical and Mental Functional Health in the European Prospective Investigation into Cancer (EPIC)-Norfolk Population-Based Study"

_jpm, 2023, doi:10.3390/jpm13091336_

Round 1

Reviewer 1 Report

This paper described an observational study on the association between HbA1c and physical/mental functional health. I find the association with MCS particularly interesting.

My only suggestion is avoiding "predictor" in the title since the study only show association but not causal relationship. 

There is minor typo on 3, line 137. HbA1 should be HbA1c.

Reviewer 2 Report

In the manuscript the predictive value of HbA1c for physical and mental function was explored in subjects with diabetes or previously unknown diabetes. The main strength of the study is the large number of included participants, but the population is divergent and nonuniform. The results of the study are somehow interesting but of limited value. I suggest several improvements before the possible publication. The comments are written below.

-The limitations of HbA1c should be described in the manuscript-which factors have an effect on the HbA1c value, that could be consequently falsely lower or higher (anemia, iron or vitamin deficiencies, renal disease, etc.). Were these factors included in the analysis? This should be added to the revised Manuscript, especially to Introduction and/or Discussion section. 

-The age range of included participants is relatively large (almost 40 years), therefore the included population is very divergent. This should also be mentioned in the manuscript.

-For the correlation of HbA1c with MCS the authors got divergent results. Which result do the authors believe to be the correct one (adjustment for covariates in model 3 or model 4)? Somehow the positive correlation between HbA1c and MCS seems not logical. How do the authors explain the results of their calculations? If the authors believe that the result is falsely positive and should be interpreted with caution (as mentioned in the Discussion) the description of the results in the Abstract should be corrected.

-What is the clinical value of the results? Which interventions do the authors suggest based on their results?

-The conclusions are missing in the manuscript.

The English language of the manuscript could be improved.

Round 2

Reviewer 2 Report

The authors corrected all remarks and the revised version of the manuscript is importantly improved.